# Plant genome sequence assembly in the era of long reads: Progress, challenges and future directions

Boas Pucker[1,2,*] , Iker Irisarri[3,4] , Jan de Vries[3,4,5] and Bo Xu[6]

[1]Department of Plant Sciences, University of Cambridge, Cambridge, United Kingdom; [2]Institute of Plant Biology & Braunschweig Integrated Centre of Systems Biology (BRICS), TU Braunschweig, Braunschweig, Germany; [3]Department of Applied Bioinformatics, Institute for Microbiology and Genetics, University of Goettingen, Göttingen, Germany; [4]Campus Institute Data Science (CIDAS), University of Goettingen, Göttingen, Germany; [5]Department of Applied Bioinformatics, Göttingen Center for Molecular Biosciences (GZMB), University of Goettingen, Göttingen, Germany; [6]State Key Laboratory of Systematic and Evolutionary Botany, Institute of Botany, Chinese Academy of Sciences, Beijing, China

## Review

**Keywords:**
haplophasing; long read sequencing; Oxford Nanopore Technologies (ONT); Pacific Biociences (PacBio); plant genome assembly; plant genomics.

**Author for correspondence:**
Boas Pucker
E-mail: b.pucker@tu-braunschweig.de

## Abstract

Third-generation long-read sequencing is transforming plant genomics. Oxford Nanopore Technologies and Pacific Biosciences are offering competing long-read sequencing technologies and enable plant scientists to investigate even large and complex plant genomes. Sequencing projects can be conducted by single research groups and sequences of smaller plant genomes can be completed within days. This also resulted in an increased investigation of genomes from multiple species in large scale to address fundamental questions associated with the origin and evolution of land plants. Increased accessibility of sequencing devices and user-friendly software allows more researchers to get involved in genomics. Current challenges are accurately resolving diploid or polyploid genome sequences and better accounting for the intra-specific diversity by switching from the use of single reference genome sequences to a pangenome graph.

## 1. Introduction

Resolving the genome structure of plants is the key to unlock the complex chassis of genetic factors determining phenotypic traits. As a biochemically homogeneous molecule, DNA can be analysed at high throughput. Enormous progress has been made in the sequencing fields over the last decades. The increase in sequencing capacity is frequently displayed outpacing Moore's law. This technological advancement facilitated major discoveries in numerous fields of life science, such as the discovery of biosynthetic gene clusters in crops (Ma, Vaistij, et al., 2021), insights into the genomic diversity of crops (Jayakodi et al., 2020; Walkowiak et al., 2020; Zhou, Chebotarov, et al., 2020), and generally a better understanding of land plant genome evolution (Carta et al., 2020; Liu et al., 2021). Plant genomics is often applied to unlock the agronomic potential of plants through identification of genetic loci underlying agronomical traits. Loci responsible for a certain trait might involve multiple genes and span hundreds or even thousands of kilobasepairs (kb). Extreme examples are biosynthetic gene clusters that can reach sizes of several hundred kb or even multiple megabases (Mbp) (Nützmann et al., 2016; Zheng et al., 2021). Therefore, it becomes useful to investigate specific allele combinations of neighbouring genes which are forming a haplotype. A sequence representing this combination of neighbouring alleles is called a haplophase. Many application cases require a genome sequence that represents all haplophases of the investigated species. Long-read sequencing is currently the method of choice to generate highly contiguous plant genome assemblies.

Here, we summarise the latest developments in the fast progressing field of plant genome sequencing, identify current challenges, highlight opportunities and postulate future directions. Our objective is to give an introduction to this field so that more plant scientists can benefit from the extensive potential of long read genomics.

## 2. Long-read sequencing technologies

There is no unified definition of "third-generation" or "long-read" sequencing technologies. Therefore, we will use a pragmatic approach and focus on the most important sequencing technologies. Refer to previous reviews about Roche/454 pyrosequencing (Metzker, 2010), Ion Torrent sequencing (Rothberg et al., 2011) or BGI's Single Tube Long Fragment Read method (Wang et al., 2019). Mainly two companies offer technologies which are expected to be the workhorses of genome sequencing projects in the future: Oxford Nanopore Technologies (ONT) and Pacific Biosciences (PacBio). The general concept and technical details of the ONT (Branton et al., 2008; Jain et al., 2017) and the PacBio (Eid et al., 2009; Hon et al., 2020; Metzker, 2010) technologies have been described and reviewed before.

Briefly, ONT sequencing is based on measuring changes of an electric signal over a membrane while a DNA strand slides through a nanopore in this membrane (Figure 1a). The recorded changes in the electric signal are characteristic for a certain composition of nucleotides partially blocking the pore and can be translated into a nucleotide sequence. Since this measuring in nanopores is not inherently restricted to DNA, this technology is currently the only method to analyse entire RNA molecules directly at high throughput. Two substantially different types of nanopores are currently distributed by ONT in the R9 and R10 flow cell families, which can be further subclassified. While R9 flow cells tend to have higher output than R10, more bases determine the signal of R10 flow cells. This is due to a longer barrel of the nanopore with a dual reader head in R10 instead of only one reader head in the R9. A reader head measures the electrical signal caused by about six bases that are located in the nanopore. Consequently, R10 flow cells are better suited to resolve homopolymers (ONT, 2021a). Models for the conversion of electric signal to a nucleotide sequence need to be trained individually for each nanopore type. An important feature of the nanopore technology is that there is no limit to the read length—other than the length/integrity of the molecule itself. The raw read accuracy can be increased from 90–95% to over 97% if a species-specific model for basecalling is available (Vereecke et al., 2020). A recent update of flow cells and chemistry enables average

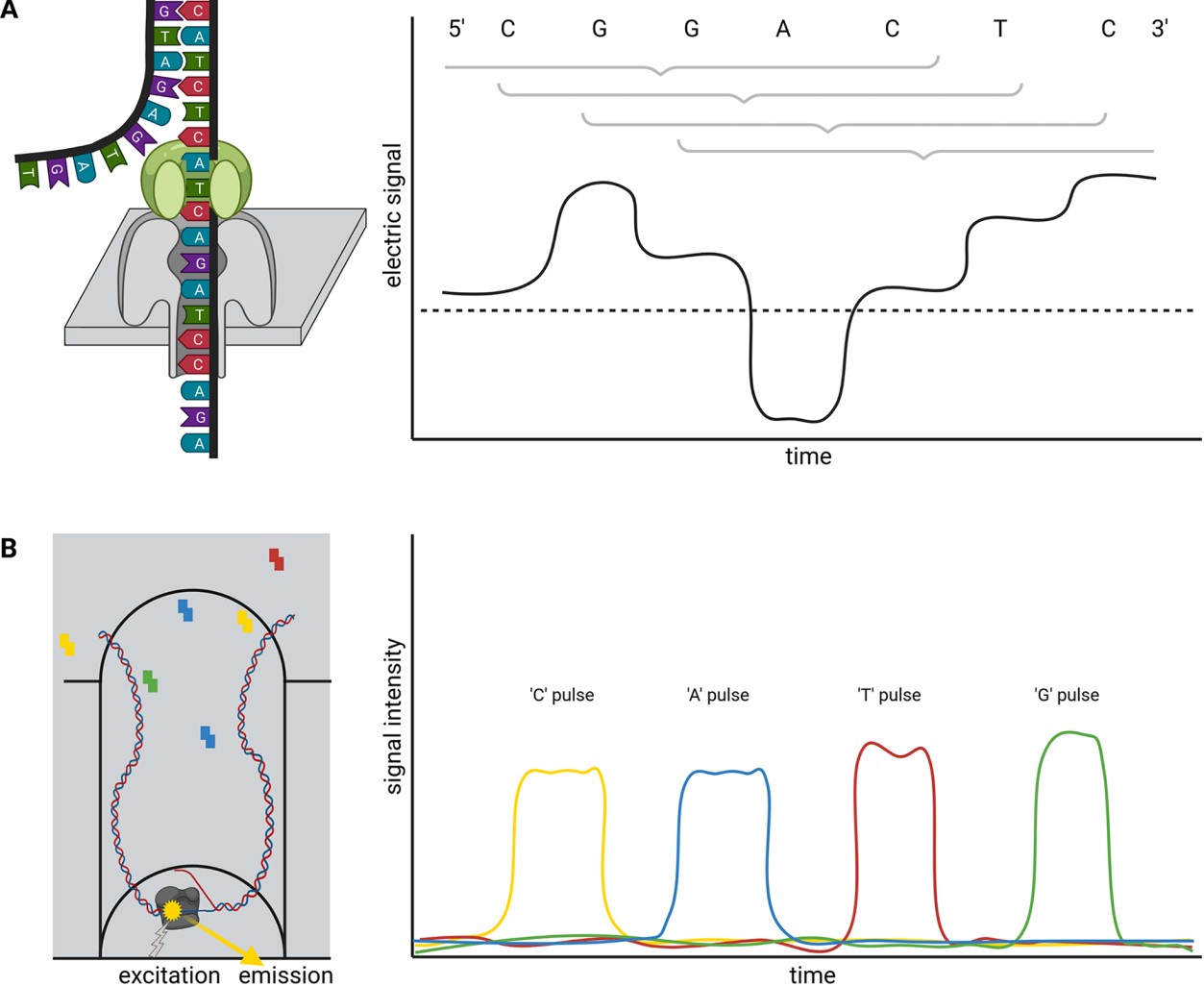

**Fig. 1.** Schematic illustration of nanopore sequencing (a) and Single-Molecule Real Time (SMRT) sequencing (b). Nanopore sequencing is based on the translocation of a DNA or RNA strand through a nanopore located in an artificial membrane. Multiple nucleotides located in the nanopore determine the flow of ions through this nanopore in a specific way by physically blocking the space. This change in ion flux is recorded as an electric signal and further converted into sequence information. The illustration shows the contribution of six bases to the signal, but the number of bases depends on the pore type. SMRT sequencing detects fluorescent light emitted from nucleotides upon incorporation into a DNA strand. The DNA polymerase is located at the bottom of a well and synthesises a new DNA strand. The integration into the new DNA strand keeps the nucleotide for a sufficiently long time in the well to allow detection.

raw read qualities around Q20 (99% accuracy). Various DNA or RNA modifications can be analysed based on ONT sequence reads (Karsten et al., 2017; Parker et al., 2019).

Single molecule real time (SMRT) sequencing offered by PacBio is based on a polymerase located in a well (Zero-Mode Waveguide, ZMW). This polymerase builds a complementary strand to a template DNA strand (Figure 1b). The incorporation of fluorescently labeled nucleotides is detected and reveals the sequence of the analysed DNA strand. PacBio offers Continuous Long Reads (CLR) and Circular Consensus Sequencing (CCS) reads also called High-Fidelity (HiFi) (Wenger et al., 2019). The later read type is the result of sequencing the same circularised DNA molecule multiple times and correcting the reads through alignment. Consequently, there is a tradeoff between the consensus read length and the per-base accuracy. The 99.5% accuracy of HiFi reads exceeds the average accuracy of CLR, but HiFi reads are usually shorter than 25 kb (Hon et al., 2020; Wenger et al., 2019). The combination of long-read length with high per-base accuracy in one technology allows the investigation of highly repetitive genomic regions.

The latest long-read technologies have the capacity to analyse extremely long DNA fragments up to millions of nucleotides in the case of ONT (Payne et al., 2019). While top read lengths of up to 500 kb can be achieved routinely in ONT sequencing runs, the longest observed plant DNA reads reached about 1.5 Mbp (Benjamin Schwessinger, personal communication). Since many sequencing projects are focussed on species without existing reference genome sequence assemblies, it is often not possible to confirm these reads through alignment against a reference genome sequence. However, long-read sequencing technologies allow to generate new assemblies for the species of interest with relative ease. Since there is no technological limit to the read length, the major challenge is the efficient isolation of high molecular weight DNA in order to obtain ultra-long reads that facilitate genome assembly. Due to the stable cell wall and a plethora of specialised metabolites, DNA extraction from plant cells is more complicated than DNA extraction from many animal cells. Challenges increase further when considering the high diversity of plants including algal species. Various DNA extraction protocols and adjustments of existing methods were developed in the last years (Li, Parris, & Saski, 2020; Siadjeu et al., 2020). Additional enrichment methods like the Short Read Eliminator kit (Circulomics) help to exclude short fragments resulting in an increased average read length. In addition to the enrichment of long molecules, reducing the amount of required DNA input is an additional challenge. Limited availability of suitable plant samples combined with large genome sizes can pose a challenge to sequencing projects. Long-read sequencing is still characterised by substantial variation between sequencing runs. This can partially be explained by differences in DNA quality. Improvements in the consumable production process might mitigate issues arising from low output runs by ensuring constant high quality. Warranty of minimal output by the supplier is a solution for the meantime. Users of commercial sequencing services might want to negotiate pricing based on the quality and quantity of sequence reads rather than on the amount of consumed materials.

## 3. High molecular weight DNA extraction for long-read sequencing

Enormous improvements of the actual sequencing capacity turned high molecular weight DNA extraction from plants into a limiting step. Many protocols for high molecular weight DNA extraction were developed previously (Jones et al., 2021; Li, Parris, & Saski, 2020; Maghini et al., 2021; Murray & Thompson, 1980; Siadjeu et al., 2020; Vilanova et al., 2020).

While the presence of long DNA molecules in the sample is crucial, short fragments can be depleted in a purification step. Moreover, the purity of the DNA is important to avoid interference with the library preparation and sequencing chemistry. Specialised metabolites and proteins might interact with the DNA and reduce the final sequencing output. Long read sequencing projects usually require several micrograms of DNA which is substantially more than needed for short-read sequencing (Siadjeu et al., 2020). This can become a challenge if no suitable plant tissues are available. Young leaves are often a good source of DNA (Pucker et al., 2021), because the number of cells (and nuclei) is high and the amount of specialised metabolites is low. Incubation in the dark for a few days can help reduce starch and sugar concentrations, thereby reducing the sugar contamination in the DNA sample. Extraction protocols should avoid shearing of the DNA molecules and storage of the final elution is recommended at 4°C. As DNA can degrade over time, the extraction should be performed in time for the sequencing experiment to ensure optimal performance.

## 4. Genome sequencing is accelerated, affordable and accessible

### 4.1. Accelerated

The 20th anniversary of the *Arabidopsis thaliana* genome sequence (Provart et al., 2020) highlights the enormous progress that has been achieved in plant genomics within two decades. While the sequencing of the first plant genome was an expensive and tedious undertaking performed by a large international consortium, *A. thaliana* genomes are now being sequenced and assembled by many labs within days (Jiao & Schneeberger, 2020; Michael et al., 2018; Pucker et al., 2019). There is also substantial progress when looking at crop genome sequencing projects. Large international genome sequencing consortia were necessary to unravel the first genome sequences of crops like rice (Goff et al., 2002; Yu et al., 2002), poplar (Tuskan et al., 2006), grapevine (Jaillon et al., 2007) and tomato (Sato et al., 2012). Now, enormous genome sequencing projects like the Darwin Tree of Life (Darwin Tree of Life Project, 2021), Earth BioGenome Project (Lewin et al., 2018) or the European Research Genome Atlas (ERGA; https://www.erga-biodiversity.eu/) are starting to sequence the genomes of all eukaryotic species within the next few years. These projects advance an open data policy and might have a positive impact beyond genomics. Therefore, it can be assumed that high-quality reference genome sequences will be available for most species in the near future. The workflow from harvesting plant material in the greenhouse or field to DNA extraction, sequencing, and *de novo* genome assembly can be completed within days (Michael et al., 2018; Pucker et al., 2021). However, current long-read technologies do not allow the construction of gapless telomere-to-telomere genome sequences on a routine basis yet. Regions like the centromere and nucleolus organising regions are not even completely resolved in the latest *A. thaliana* genome assemblies (Michael et al., 2018; Pucker et al., 2021). Consequently, challenges to close the remaining gaps in genome sequences of most species will remain for the foreseeable future. Since the read lengths of both long-read technologies is impressive, the major factor to optimise in the future is per-base accuracy. Rapid increase of the raw read quality during the last years accelerated many genome

| | task | consumed time | hands-on time | equipment | estimated costs of consumables | estimated costs of lab equipment |
|---|---|---|---|---|---|---|
| A | plant incubation in darkness | 2-3d | 1h | | | |
| B | non-destructive sampling | - | 1h | | | |
| C | DNA extraction | 1d | 8h | waterbath, centrifuge | $50 | $1000 $8000 |
| D | quality control | 1h | 1h | NanoDrop, Qubit | $20 | |
| E | short fragment depletion | 2h | 1h | centrifuge | $50 | |
| F | quality control | 1h | 1h | NanoDrop, Qubit | $20 | $5000 $5000 |
| G | library preparation & sequencing | 1-5d | 4-16h | centrifuge, magnetic rack, sequencer | $3000 | $250 $1000 |
| H | basecalling | 1d | 1h | computer with GPU | | $3000 |
| I | assembly | 1-15d | 1h | | | |
| J | polishing | 1-5d | 1h | compute cluster / cloud | | |
| K | annotation | 1-5d | 1h | | | |
| L | data submission | 2h | 2h | fast internet connection | | |

**Fig. 2.** Plant genome project workflow from DNA extraction over Oxford Nanopore Technologies (ONT) sequencing to data submission. The indicated durations depend on the size and complexity of the investigated plant genome, with larger genomes generally taking longer to analyse. To reduce sugar content, plants are incubated in the dark for a few days prior to DNA extraction (a). Non-destructive sampling is important to allow additional genomic sequencing and also RNA-Seq if required in later stages of a project (b). Mechanical disruption of cell walls is required for the DNA extraction (c). Photometric analysis of the DNA solution (including quantification) is often the first step of quality control (d and f). Removal of short DNA fragments is highly recommended to improve the sequencing output and quality (e). ONT library preparation and sequencing can be repeated several times to increase the output (g). Graphic cards are an efficient resource to convert electric signal into sequence information in real time (h). Multiple tools are available to generate a chromosome-arm level assembly based on long reads (i). Additional polishing in multiple rounds can be necessary due to the noisy character of long reads (j). The value of a genome sequence can be enriched through the identification of relevant genetic elements like genes and transposable elements (k). All data should be shared with the community via submission to a public repository which ensures long-term storage (l). d, day(s); hr, hour(s). The given time estimates for assembly, polishing and annotation are the minimal run time required for the analyses. Manual curation and iterative improvements can take substantially longer. The estimated costs of consumables are based on a haploid 1-Gbp genome and a targeted coverage of 30× which would require six libraries to be sequenced on three MinION/GridION flow cells when assuming an average output of 10 GB per flow cell with two libraries sequenced per flow cell. Investment costs for non-standard lab equipment are independent of the specific sequencing project and only required for high-output experiments in the lab. There is an option to perform rapid sequencing without these instruments in the field, but the lower output does not make that option attractive for large plant genomes.

sequencing projects. PacBio offers HiFi reads which are highly accurate and up to 25 kb long. Since per-base accuracy is based on sequencing the same molecule numerous times, improving the polymerase lifetime could increase raw read accuracy and simultaneously shift the length limit. ONT recently released a 'Q20+' technology together with R10.4 flow cells, which is pushing the raw read accuracy beyond 99% (ONT, 2021b). Since the length of ONT reads is only limited by the length of the DNA molecule, this could become the routine technology to resolve rDNA clusters. The high accuracy of PacBio and ONT long reads accelerates the assembly process and removes the need for short-read polishing, which was previously required to correct errors in non-repetitive regions. As short reads cannot be mapped onto sequences of repetitive regions with reliability, long-read only assemblies could also accelerate the research on transposable elements.

### 4.2. Affordable

The distribution of affordable ONT MinION sequencers started the democratisation of sequencing (The long view on sequenc-

ing, 2018). Increase in read length and output enabled substantial improvements of assembly contiguity and reduced costs associated with genome sequencing projects. Genome sequencing is likely to replace classic polymerase chain reaction-based genotyping methods in certain application cases due to higher cost-effectiveness (Pucker et al., 2021). Plant genome assemblies at chromosome-arm level often cost less than $10,000 and can be completed within days to weeks for many species (Figure 2). However, reaching a telomere-to-telomere assembly is still difficult and expensive. Commercial service centres offer the generation of data at continuously decreasing prices rendering genome sequencing affordable for most research groups. This democratisation might shift the focus of genome sequencing projects from crops with importance in agriculture to neglected crops in developing countries. Improved technologies and substantially reduced sequencing costs have the potential to establish genome sequences as a standard for all plant species. Genetic markers, Hi-C or optical mapping data can be used to arrange contigs into representations of entire chromosomes so-called pseudochromosomes or C-scaffolds (Lewin et al., 2019; Li, Xiang, et al., 2020; Paajanen et al., 2019). Pseudochromosomes

contain ordered contigs connected by stretches of ambiguous bases (Ns) to indicate assembly gaps that are only bridged by information about the distance of specific sequences without knowledge about the interleaved sequence. The concept could be considered analogous to paired-end or mate-pair reads, but the distance between the markers is substantially larger. Assemblies generated with the latest long-read technologies can surpass long-standing reference genome sequences with respect to quality and contiguity (Pucker et al., 2021; Rai et al., 2021). Portable sequencers like MinION and Flongle might not be the choice for crop genome sequencing because affordability and throughput are more important than on-site sequencing.

### 4.3. Accessible

Initial crop genome sequencing projects relied mostly on short reads of second-generation sequencing technologies such as Roche/454 pyrosequencing and Illumina sequencing-by-synthesis which are only accessible to large sequencing centres that can afford the maintenance of expensive instruments. Costs associated with PacBio sequencers still prevent single research groups from buying their own instruments; thus services provided by companies or core facilities are required. However, portable ONT sequencers provide new opportunities for small labs thereby opening an unprecedented opportunity for genome sequencing in low-income countries and for non-model plants such as algae. Substantially, more researchers get involved in genome sequencing and the awareness for opportunities increases. It is also likely that orphan crops, that is species with untapped economic potential, will be made accessible through the publication of their genome sequences (Hunt et al., 2020; Siadjeu et al., 2020; Wang et al., 2021). Huge community engagement inspired the development of more user-friendly and mobile software tools (de Koning et al., 2020; Oliva et al., 2020; Palatnick et al., 2020; Samarakoon et al., 2020), which are paving the way for the democratisation of sequencing data analysis. Both PacBio and ONT come with the opportunity to

identify DNA modifications. Even if this opportunity is not used in all sequencing projects, re-use of datasets is possible if all raw data are deposited in public repositories like the Sequence Read Archive and European Nucleotide Archive. Pure bioinformatics groups without experience in genome sequencing can harness these datasets for their analyses. Finally, there is also an educational aspect to portable sequencers. MinION and Flongle can be used to perform plant genomics projects in practical courses at universities and beyond. Persons with basic laboratory skills can operate these sequencers based on instruction videos and manuals without additional training.

### 5. Pangenomics: From re-sequencing to reference quality genome assemblies of cultivars

The pangenome concept describes all genes or more generally genetic information that is present in a certain group of individuals, for example a population, a species or a higher taxonomic unit. Pangenomes comprise a small set of essential or core genes and numerous genes with different levels of dispensability some of which might be 'accessory' genes (Marroni et al., 2014; Sielemann et al., 2021). A single assembly cannot capture the complete set of genes present in a species and thus a species' pangenome is a better reflection of the diversity. In plants, accessory genes are often enriched in functions related to biotic and abiotic stress response (Bayer et al., 2020). The objective of earlier genome sequencing consortia has been to construct one reference genome sequence that would not just benefit research on one particular species, but would also support research on related species. In such cases, variations in different cultivars or related species were investigated by short read-based re-sequencing and mapping to the reference genome sequence (Figure 3). For example, such studies investigated the pangenome of the model species *A. thaliana* (Alonso-Blanco et al., 2016), tomato (Causse et al., 2013), rice (Lv et al., 2020) and grapevine (Liang et al., 2019). Despite their success, such short-read re-sequencing projects have inherent limitations such

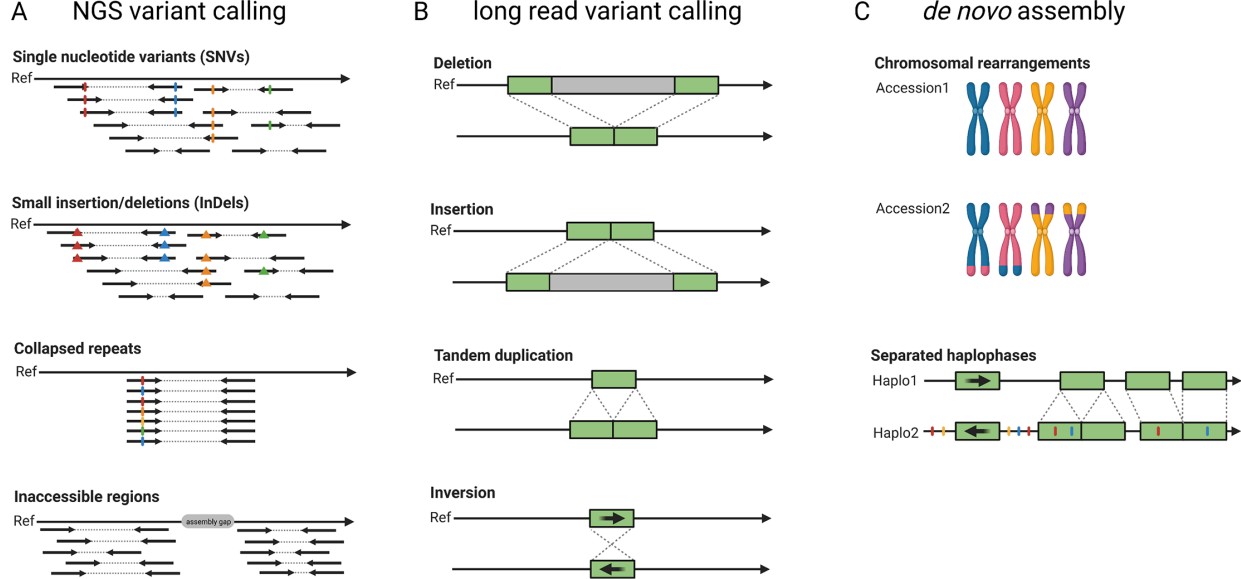

**Fig. 3.** Development of sequence analysis for exploring genome structure and variability. Read mapping and variant calling was the initial approach to characterise differences between samples based on short-read ('NGS') data (a). Long reads allow an improved variant detection which is especially suited for the detection of structural variants (b). Independent *de novo* genome assemblies allow the identification of all variants and already include an assignment of variants to haplophases (c).

as the inabilities to resolve large insertions or to identify variants in repetitive or heterozygous regions (Cameron et al., 2019; Schilbert et al., 2020). Long reads enable the identification of structural sequence variants which have not been identified based on short reads (Chawla et al., 2021). The detection of single nucleotide variants (SNV) requires dedicated tools like DeepVariant (Poplin et al., 2018) and LongShot (Edge & Bansal, 2019), but can outperform the SNV detection based on short reads in difficult-to-map regions (Olson et al., 2021). Nevertheless, *de novo* genome assemblies for multiple different cultivars and comparison of the resulting genome sequences is likely to replace classic variant calling against one reference sequence in most applications (Bayer et al., 2020; Michael & VanBuren, 2020). The feasibility and advantages of constructing a *de novo* genome assembly for the discovery of sequence differences within one species were demonstrated for *A. thaliana* (Michael et al., 2018; Pucker et al., 2021). First crop genome projects generated independent long-read assemblies covering the crop species rice (Choi et al., 2020; Stein et al., 2018; Zhou, Chebotarov, et al., 2020), rapeseed (Song et al., 2020), apple (Sun et al., 2020), wheat (Walkowiak et al., 2020), barley (Jayakodi et al., 2020), soybean (Liu et al., 2020), sorghum (Tao et al., 2021) and maize (Hufford et al., 2021). These studies identified large structural variants including translocations, insertions, deletions, inversions and chromosome fusions. They also found that some 'accessory' genes can have large phenotypic effects including ecotype differentiation, flowering time, stress tolerance or seed weight (Song et al., 2020; Walkowiak et al., 2020). Additional genes and other cultivar specific sequences can be discovered in these projects, but the study of pangenomes should not be limited to crops, because wild relatives might harbour a richer set of 'accessory' genes (Bayer et al., 2020). Some of these 'accessory' genes (e.g., pathogen resistances) could be introduced into crops through breeding. Clearly, long-read sequencing plays a crucial role in the transition towards plant pangenomics.

## 6. Understanding the deep roots of plant evolution through genomics

Comparative genomic analyses of land plants and their algal relatives provide an unprecedented opportunity to investigate the origin and evolution of embryophytes and their traits. Some agriculturally relevant traits such as tolerance of water scarcity and mutualistic symbioses have deep evolutionary origins predating the origin of land plants. Insights into the evolution of these traits is not only relevant for the understanding of plant terrestrialisation, but can thus also have agronomical implications (Bowles et al., 2021; Fürst-Jansen et al., 2020). Complemented with functional genomic studies, comparative genomics shed light on the innovations of land plant traits such as water conduction systems (Xu et al., 2014), rooting systems (Menand et al., 2007), membrane modifications (Resemann et al., 2021), cuticle (Xu et al., 2021) and stomata (Chater et al., 2017). Deciphering the genomes of species occupying critical phylogenetic positions revealed information on the origin and early evolution of seed-free plants (Szövényi et al., 2021), gymnosperms (Liu et al., 2021), flowering plants (Zhang, Chen, et al., 2020), and grasses (Ma, Liu, et al., 2021), and the genomes of land plants' algal relatives provide a better understanding of genetic changes underpinning the water-to-land transition and associated stress adaptations (Cheng et al., 2019; Jiao et al., 2020; Nishiyama et al., 2018; Wang et al., 2020). In fact, 'alga' is a general term for photosynthetic eukaryotes (and historically also cyanobacteria), which include not only streptophyte algae and land plants, but also an astonishing diversity of green, red and glaucophyte algae—all of which are derived from the singular primary endosymbiotic incorporation of the cyanobacterial progenitor of plastids (de Vries & Archibald, 2017; Keeling, 2013; Sibbald & Archibald, 2020). Additionally, many other eukaryotic groups secondarily acquired plastids by eukaryote–eukaryote endosymbioses, including brown and golden algae or diatoms, among many others (Keeling, 2013; Sibbald & Archibald, 2020; Strassert et al., 2021). This long and convoluted evolutionary history translates into an extraordinary diversity of genomes (Blaby-Haas & Merchant, 2019). Interpretation of these genomes has important biological and biotechnological implications. Over 100 algal genomes have been sequenced to date (Grigoriev et al., 2021) and more are to come.

Until recently, very few algal genome sequences could be considered complete (telomere-to-telomere) and these were on the small range of genome size, with most other assemblies having variable completeness from very short contigs to chromosome-level assemblies (Blaby-Haas & Merchant, 2019). Given the high phylogenetic diversity of algae and the fact that specimens are often sourced from natural populations (most are non-model organisms), high heterozygosity and the presence of many repetitive elements can hamper the assembly of a high-quality algal genome sequence (Michael & VanBuren, 2020). With the exception of a few algal model systems such as *Chlamydomonas reinhardtii* (O'Donnell et al., 2020), *Cyanophora paradoxa* (Price et al., 2019), *Phaeodactylum tricornutum* (Filloramo et al., 2021) and *Thalassiosira pseudonana* (Armbrust et al., 2004), most algal genomes are relatively poorly characterised in comparison to flowering plants. Fortunately, new algal models are flourishing, be it *Ectocarpus siliculosus* (Coelho et al., 2012), *Nannochloropsis* spp. (Radakovits et al., 2012) or *Ulva mutabilis* (De Clerck et al., 2018). A list of available algal and non-seed plant genomes is shown in Table 1. As in other non-model organisms, functional annotation of algal genomes is hampered by the large phylogenetic distance to current model species in which proteins have been functionally characterised (often flowering plants). The likelihood of finding orthologs with the same function across long evolutionary times is low. Currently, about half of the annotated proteins in algal genome sequences, on average, lack functional annotation obtained by searches against Pfam or EggNOG databases (Blaby-Haas & Merchant, 2019). This suggests that algae harbour a vast genetic potential and new gene functions that are yet to be discovered through biochemical characterisation. Gene family analysis using protein similarity networks, co-expression networks and phylogenetic reconstruction are powerful methods to improve functional annotation, providing information on protein domains, condition-specific gene regulation and evolutionary links from knowns to unknowns (de Vries et al., 2021; Gong & Han, 2021; Li et al., 2015; Rhee & Mutwil, 2014; Ruprecht et al., 2017)—especially when novel lineages of algae are involved (Li, Wang, et al., 2020). Reliable genome sequences are the foundation for all these approaches.

Besides nuclear genomes, the plastid (plastome) and mitochondrial (chondrome) counterparts are often of interest in evolutionary biology. The automatic generation of full plastid or mitochondrial genome sequences is now possible as a byproduct of nuclear genome sequencing projects. Long reads also make the more complex chondrome more accessible to genomic studies. Various pipelines have been implemented for the assembly of organellar genomes using exclusively long-read or in combination with short-read data (Soorni et al., 2017; Wick et al., 2017).

**Table 1.** Available streptopohyte algae and non-seed plant genomes salient to our understanding of plant diversity and evolution

| Species | Assembly size (Mb) | Scaffold N50 (bp) | Lineage | Reference |
|---|---|---|---|---|
| *Chara braunii* | 1,751.21 | 2,261,426 | Streptophyte algae (Charophyceae) | Nishiyama et al., 2018 |
| *Chlorokybus atmophyticus* | 74.33 | 752,385 | Streptophyte algae (Chlorokybophyceae) | Wang et al., 2020 |
| *Klebsormidium nitens* | 104.21 | 134,930 | Streptophyte algae (Klebsormidiophyceae) | Hori et al., 2014 |
| *Mesostigma viride* | 441.7 | 2,558,729 | Streptophyte algae (Mesostigmatophyceae) | Wang et al., 2020 |
| *Mesotaenium endlicherianum* | 173.75 | 448,375 | Streptophyte algae (Zygnematophyceae) | Cheng et al., 2019 |
| *Penium margaritaceum* | 3,661 | 116,100 | Streptophyte algae (Zygnematophyceae) | Jiao et al., 2020 |
| *Spirogloea muscicola* | 170.82 | 566,364 | Streptophyte algae (Zygnematophyceae) | Cheng et al., 2019 |
| *Anthoceros agrestis (Bonn)* | 116.9 | 17,300,000 | Hornworts | Li, Nishiyama, et al., 2020 |
| *Anthoceros angustus* | 119.35 | 796,643 | Hornworts | Zhang, Fu, et al., 2020 |
| *Anthoceros punctatus* | 132.8 | 1,700,00 | Hornworts | Li, Nishiyama, et al., 2020 |
| *Marchantia inflexa* | 208.75 | 11,136 | Liverworts | Marks et al., 2019 |
| *Marchantia paleacea* | 250.80 | 2,390,877 | Liverworts | Radhakrishnan et al., 2020 |
| *Marchantia polymorpha* | 225.76 | 1,366,373 | Liverworts | Bowman et al., 2017 |
| *Ceratodon purpureus* | 362.51 | 1,405,213 | Mosses | Carey et al., 2021 |
| *Fontinatis antipyretica* | 385.2 | 45,800 | Mosses | Yu et al., 2020 |
| *Funaria hygrometrica* | 340 | 100,000 | Mosses | Kirbis et al., 2020 |
| *Pleurozium schreberi* | 318.34 | 154,439 | Mosses | Pederson et al., 2019 |
| *Physcomitrium patens* | 472.08 | 17,435,539 | Mosses | Lang et al., 2018 |
| *Sphagnum fallax* | 395,1 | 21,100,000 | Mosses | *Sphagnum fallax* v1.1, DOE-JGI, http://phytozome.jgi.doe.gov/ |
| *Sphagnum magellanicum* | 439.0 | 23,200,000 | Mosses | *Sphagnum magellanicum* v1.1, DOE-JGI, http://phytozome.jgi.doe.gov/ |
| *Syntrichia caninervis* | 329.82 | 21,898,694 | Mosses | Silva et al., 2021 |
| *Isoetes taiwanensis* | 1,660 | 17,400,000 | Lycophytes | Wickell et al., 2021 |
| *Selaginella lepidophylla* | 122 | 163,000 | Lycophytes | VanBuren et al., 2018 |
| *Selaginella moellendorffii* | 212.31 | 119,796 | Lycophytes | Banks et al., 2011 |
| *Selaginella tamariscina* | 300.73 | 407,666 | Lycophytes | Xu et al., 2018 |
| *Azolla filiculoides* | 750 | 964,700 | Ferns | Li et al., 2018 |
| *Ceratopteris richardii* | 7,462.46 | 2,273,607 | Ferns | Marchant et al., 2019 |
| *Salvinia cucullata* | 260 | 719,800 | Ferns | Li et al., 2018 |

Denoted are numbers on the total assembly size, contiguity statistics (N50), taxonomic affiliation and references. Genome statistics were obtained from NCBI's Assembly database or the corresponding publications.

## 7. From haploid to diploid genome assembly

Crop genome sequencing projects were focussed on almost homozygous cultivars (Jaillon et al., 2007) or even doubled haploid lines when possible (Dohm et al., 2014). Even human genome initiatives that are usually a few years ahead of plant sciences, have only recently managed to produce a complete haploid genome assembly (Nurk et al., 2021). This implies that two separate genome sequences need to be assembled to represent the two haplotypes of heterozygous genotypes. Haplotypes are the biological molecules i. e. a group of alleles that are inherited together. Haplotypes are represented by haplophases in the assembly. The need to distinguish between these two haplophases when targeting heterozygous genotypes adds an additional overhead that makes the situation more complicated. When possible, genome sequencing projects avoided the challenge of separating haplophases by focussing on homozygous or haploid genotypes. The genomes of polyploidy species are an even bigger challenge, because more than two haplotypes need to be represented in the assembly. Polyploid

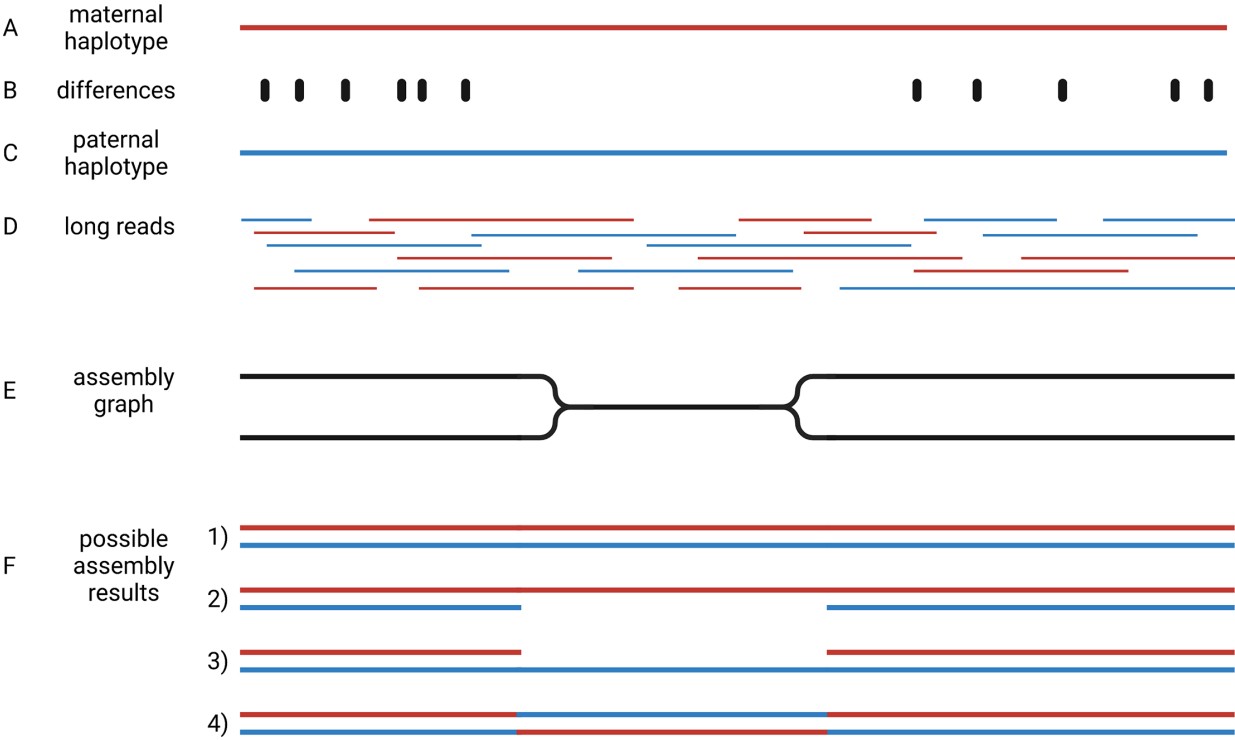

**Fig. 4.** Assembly of haplophases. Diploid plant genomes have a maternal (a) and a paternal haplotype (c), which differ at specific positions (b). Long reads belong to one or the other haplotype (d). The assembly graph separates haplophases in regions with sufficient differences between both parental haplotypes, but collapses them in identical (homozygous) regions (e). Resolving the assembly graph into final sequences is possible in four different ways (f): It is possible that both haplophases are resolved by connecting the two divergent blocks correctly (1), identical regions could be assigned to one haplophase leading to a less continuous second haplophase (2 and 3), or the identical region can cause an erroneous connection of the flanking distinct sequences (4). This illustration shows the analysis of a diploid genome, but the concept is generalisable to polyploids.

species were investigated by sequencing relatives with a lower ploidy (Kyriakidou et al., 2018; Schmutz et al., 2010; Zimin et al., 2017). Substantially increased read length and sequence accuracy of ONT and PacBio reads enabled the investigation of more challenging genomes. Objectives of current genome sequencing projects are the construction of phased genome sequences that represent both haplotypes by accurate haplophases (Girollet et al., 2019; Siadjeu et al., 2020; Sun et al., 2020). Accurate separation of haplophases is particularly important in highly heterozygous species like grapevine, in which alleles can differ by numerous presence/absence variations. While several assemblies of heterozygous species contain contigs representing two haplophases, it is not clear if contigs accurately represent a single haplotype. One major challenge to the accurate assembly of haplophases is the heterogeneous distribution of differences between the haplotypes. Regions rich in differences are easily separated into phases, but such regions are interleaved with homozygous regions that are more difficult to separate (Figure 4). The major challenge is to avoid switches between the haplophases in these homozygous regions.

Incorporation of external information, for example parental sequencing data are well-established approach to separate haplophases. TrioBinning identifies unique k-mers in each of the parental sequencing datasets and bins the reads of their offspring accordingly (Koren et al., 2018). This approach allows the separate assembly of both haplophases, avoiding phase-switching issues. Each assembly is resolving the structure of one haploid genome. Other approaches subject gametes to single cell sequencing (Campoy et al., 2020; Shi et al., 2019) because these cells contain only DNA of one of the haplotypes. The availability of HiFi reads enables the accurate assembly of haplophases (Zhou, Tang, et al., 2020). Another approach is based on high-throughput chromosome conformation capture (Hi-C) or Omni-C (Dovetail Genomics) data, which provide information about the physical proximity of different parts of the chromatin. Briefly, DNA strands are cross-linked with formaldehyde and digested by endonucleases. Cross-linked DNA fragments are ligated with an adapter in between and subjected to sequencing. It is similar to a mate pair library with huge insert sizes. Tools like ALLHiC (Zhang et al., 2019), hifiasm (Cheng et al., 2021) and FALCON-phase (Kronenberg et al., 2021) allow the integration of these data for a high-level scaffolding of large contigs in an allele-specific manner, thus paving the way for phased assemblies of heterozygous and polyploidy species.

## 8. Computational future of plant genomics

While sequencing costs drop and the amounts of data increase, the computational data analysis has become the major challenge. Higher raw-read accuracy is likely going to change this again, but the conversion of physical signals into sequence information (basecalling) during the actual sequencing process remains computationally intensive. For example, ONT's GridION is relying on the processors of graphic cards to perform basecalling in real time. Performing the basecalling after completion of a sequencing run on CPUs is an alternative, for example when using a MinION. Miles Benton maintains an excellent blog about technical details and gives advices about the best graphic cards for basecalling of ONT data (Benton, 2021). The primary analysis of PacBio sequencing

data involves multiple steps resulting in trace, pulse, base and FASTQ or BAM files. The base file is usually stored as it provides the basis for all secondary analyses. In contrast to second generation short-read sequencing technologies, it is important to store raw data (fast5 for ONT and base files for PacBio) of long-read sequencing runs. Rapid improvements in the basecalling algorithms (Amarasinghe et al., 2020) will allow drawing substantially more accurate information from the same raw reads in the future. The rapid development of new basecalling tools also poses a challenge to users looking for the best solution.

Genome assemblies based on noisy long reads often require a first correction step, which involves the computationally challenging all-vs-all alignment of reads. This step involves the generation of temporary files which are several times the size of the initial sequence data (FASTQ files). More stringent settings in the detection of matches between the reads can help reduce the disk space requirements in this step. The >99.5% accuracy of HiFi reads is a first step to reduce the computational costs of plant genome assemblies by an order of magnitude (Cheng et al., 2021; Mascher et al., 2021; Nurk et al., 2020) because alignments between reads can be restricted to almost perfect matches or the correction step can be skipped altogether.

Genome assemblies require high-performance hardware. However, their usage is characterised by peaks in memory and CPU consumption for assemblies and idle time while no assemblies are computed. Institutional compute clusters can make the necessary resources available to users for the assembly process, but not all institutions can offer this support. Commercial cloud computing offering large resources temporarily could be a good solution for groups that do not have access to high-performance hardware. However, data storage and transfer remains expensive. Several organisations already recognised this issue and offer computational resources and support for researchers, for example de.NBI (Belmann et al., 2019) and CYVERSE (https://cyverse.org/). As described for basecalling and read correction, the settings of the assembly process influence the required computational resources. There is a trade-off between the quality of a genome representation and the associated computational costs (Kaye & Wasserman, 2021). (Hi)Canu (Nurk et al., 2020; Zimin et al., 2017) produced the plant genome assembly of choice in many projects, but other assemblers like Flye (Kolmogorov et al., 2019) might be better if repetitive sequences are the focus of a study (Naish et al., 2021).

While genome assemblies are 'only' computationally challenging, the prediction of gene models and the functional annotation of predicted gene models will remain a challenge for the foreseeable future. The prediction of gene models is usually supported by RNA-Seq. The direct RNA sequencing offered by ONT or full length cDNA sequencing by PacBio or ONT is a good way to improve the annotation and detection of splicing isoforms. Given that multiple genome sequences of closely related plants are generated, the identification of gene models should be performed simultaneously on all sequences as implemented in the Comparative Annotation Toolkit (Fiddes et al., 2018). However, there are many other tools or pipelines including BRAKER2 (Brůna, Hoff, et al., 2020; Hoff et al., 2019), SNAP (Korf, 2004), GeneMark-EP+ (Brůna, Lomsadze, & Borodovsky, 2020) and Gnomon (Souvorov et al., 2018).

Many different tools for the analysis of long-read data are available and new ones are continuously developed. Every tool has its specific strengths and weaknesses with respect to applications, but this also depends on the nature of the data at hand. Therefore, there is a need for benchmarking studies to provide guidance to potential users. Benchmarking studies on short-read assemblers like the Assemblathons (Bradnam et al., 2013; Earl et al., 2011) were informative for many years until long-read sequencing technologies became the *de facto* standard for plant genome assemblies. However, a mechanism to continuously update the benchmarking results would be important for modern long-read assemblers. New software and technology versions are frequently released, thus making comparisons obsolete within months. There are efforts to optimise assemblers towards speed and reduced memory usage (Gatter et al., 2021; Haghshenas et al., 2020; Shafin et al., 2020). While this is important to complete extremely large plant genome assemblies and to reduce the environmental impact of bioinformatics, quality improvements are still of interest and would be beneficial for smaller genomes. Projects aiming for better assembly quality are often trying to achieve this through accurate separation of the haplophases (Chin et al., 2016; Koren et al., 2018; Nurk et al., 2020).

## 9. Conclusion

Genome sequencing is a rapidly developing field with an exponential growth in the amount of produced data and biological insights gained from them. Technological developments solve the longstanding assembly contiguity issue and enable novel analyses like the study of DNA modifications at a genome-wide scale. As a consequence, we as genomicists gain not only quantity, but also quality. The accurate separation of haplophases remains a challenge. Open science principles including an effective data sharing have been important in the past and will open even more opportunities in the future. Dropping sequencing costs and technological improvements will help to move from single reference genome sequences to pangenomics in order to better understand the genomic diversity within every species.

## Acknowledgements

We thank Quantitative Plant Biology for the invitation to submit this review article. Some figures were generated using bioRender.com.

**Financial support.** B.P. is funded by the Deutsche Forschungsgemeinschaft (DFG, German Research Foundation)—436841671. I.I. and J.d.V. are part of the framework of MAdLand (http://madland.science, DFG priority programme 2237); J.d.V. is grateful for funding by the DFG (VR132/4-1). Work in the lab of J.d.V. is further supported by funding from the European Research Council (ERC) under the European Union's Horizon 2020 research and innovation programme (grant agreement no. 852725; ERC Starting Grant 'TerreStriAL'). B.X. is supported by the National Natural Science Foundation of China (32070249), and the Strategic Priority Research Programme of the Chinese Academy of Sciences (XDA26030104).

**Conflicts of interest.** B.P. was an invited speaker without financial compensation at a virtual conference (London Calling 2021) organised by Oxford Nanopore Technologies. J.V., I.I. and B.X. declare no conflicts of interest.

**Authorship contributions.** B.P. initiated and coordinated the project. All authors contributed to the manuscript and have approved the final version.

**Data and availability statement.** Data availability is not applicable to this article as no new data were created or analysed in this study.

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
