## [Reviewer Report]

Dear Editor,

We are submitting our paper "Plant genome sequence assembly 3.0: progress, challenges, and future directions" by Boas Pucker, Iker Irisarri, Jan de Vries and Bo Xu for publication in Quantitative Plant Biology. This review article was invited by the Editor-in-Chief, Prof. Olivier Hamant.

Our review deals with the latest developments in long read sequencing technologies and their implications on plant genome assembly. Highlights are the increased availability of sequencing instruments, the reduced time and cost for plant genome projects, and the development towards pangenome projects. Since many of the latest insights into the evolution of land plants are based on algal genome sequencing projects, we also review the development in this rapidly developing field.

We hope that you will find our manuscript suitable for publication in Quantitative Plant Biology and hereby confirm that the paper is not under consideration for publication elsewhere. We declare that all potential conflicts of interests are declared. All authors have approved the manuscript and agree with its submission to Quantitative Plant Biology.

Best regards,

Boas Pucker

(on behalf of all authors)

---

## [Reviewer Report]

*Comments to Author*: The authors have pulled together a comprehensive review manuscript that addresses the current state of long-read sequencing with PacBio and ONT technologies, their limitations, opportunities, and the future of the field as a whole. In general, some sections would benefit from more direction and clarity, as ultimately the major message about the power and opportunities for the future of genomics was sometimes lost. I have comments about a few of the sections below that I hope are helpful.

Section Long read sequencing technologies: This was a nice overview of two major long-read technologies. My one comment is that a casual reader of the first section of “long read sequencing technologies” would likely get the impression that the error rates for ONT are <3% and the error rate for HiFi < 0.5%. There is more nuance here that is left out. The high-level tradeoff that would be helpful to more explicitly say is that HiFi reads are medium-sized but close to perfect, and you need access to a sequencing core, but the compute power for assembly is reduced…whereas nanopore reads can be much longer but suffer from higher error rates — and typically much higher than 3% error — but you can do it yourself, although the all-by-all read correction for assembly is still computationally costly. A more explicit description of the error rate with nanopore reads would be helpful. 

Section “Genome sequencing is accelerated, affordable, and accessible”: This section would benefit from a revision that more clearly addresses those 3 words — accelerated, affordable, and accessible. Right now, this section is a single paragraph that addresses the history of genome sequencing, ONT prices, democratization, Hi-C sequencing, orphan crops, scaffolding techniques, back to democratization and small labs, and then bottlenecks in HMW isolation, then educational aspects. This paragraph that would benefit from subheaders or, at the very least, more clearly defined paragraphs that address a major theme — and I completely agree that “quick, cheap, and everywhere” are 3 excellent points to highlight, that each deserve their own paragraphs.

Figure 2 describes some highly optimistic timelines. 1 hands-on hour for genome assembly is not realistic. Same for polishing, and especially the same for annotation. Building custom repeat databases, and setting up Maker-P runs is still a painful ordeal. It misrepresents the amount of work that goes into producing a polished, scaffolded genome, especially in larger plant genomes. I agree that running the software can sometimes be quick, but there is more work that goes into ensuring high quality assembly and annotation, as it is often an iterative process. This kind of figure gives a reader the impression that assembly and annotation is a solved problem, when I would argue that there is still an immense amount of work that goes into ensuring high quality assemblies and annotations before release. I would also specify in the figure legend that this workflow is specific to ONT. 

Section “Understanding the deep roots of plant evolution”

This section does not necessarily fit the flow of the manuscript, as it intensely focuses on algae as an understudied group, and describing interesting attributes about algae. Although there is little information about genomics or genome assembly here, except as a concluding sentence L252 “Over 100 algal genomes have been sequenced to date, and more are to come”, I agree completely with the authors’ call to arms that we need more intense focus on the genomics of algae.

Section From haploid to diploid genome assembly: My only suggestion with this section is to include some detail about how Hi-C/Omni-C can also be useful for extending haplotype phase blocks, e.g. FALCON-Phase and hifiasm’s new hi-c integration mode. This is a very nice figure in this section that clearly describes the problem(s).

Section “Genome sequencing and assembly - a dead end?”

I would suggest restructuring the last section on whether genome assembly is a dead end, as it seems to defeat the major message of the manuscript. The authors describe a scenario where a few mega-genome projects will quickly eliminate the need for anyone else to generate a genome assembly for a plant. I have high, and reasonable, skepticism that reference genomes for all eukaryotes will be available in the near future. The authors have already made the point (L195) that a single genome assembly for a plant does not capture adequate diversity of a species, so it seems that is the major point that could be made in this section: long-read sequencing is poised to continue to drop in price, accessibility, and ease, so communities of scientists will be able to extend the reach of these larger, global, single-genome-per-species projects by expanding outward from just a single reference per species. That sentiment ties together much of the manuscript, that sequencing is cheap and accessible now, and that species pangenomes are necessary for capturing the diversity of a species. This sentiment is somewhat mentioned (L402), but I was still left deflated and thinking the authors believe that the future of genomics should be left in the hands of just a few mega-projects, which I don’t believe the authors intended.

The last two paragraphs of this section drift into unclear territory and don’t contribute to the main thread of this section. 

Minor comments:

L82: I would buffer this statement and connect it back to the need for model training. In general, if we sequence a random plant genome on nanopore, we are getting ~10% error rates. 

L86: Zero-Mode Waveguide (ZMW)

L148: This sentence set up two different bottlenecks but never discusses them.

L338: I don’t agree that this is generally true. Yes, there are many fast assemblers being developed, but there is still plenty of focus on Hi-C integrated builds, trio bins, and fully phased haplotype-aware assembly (e.g. hifiasm, HiCanu).

L354: Is there data to support this claim that most genome projects focus on novel species?

L409: This paragraph does not fit the flow of the section and could be moved to the assembly section.

---

## [Reviewer Report]

*Comments to Author*: The invited review of Pucker et al. is a summary of recent progress in sequencing technology. The review is well-written. The relevant facts have been selected and they have been presented in a correct way. I’m uncertain though for whom this review is written: advice on how to negotiate with sequencing providers is targeted to rookies; the last “dead end” paragraph reminds silverbacks to look for new research goals now that genome assembly is a piece of cake. An explicit mission statement may help.

Here are some minor suggestions for improvement. They are mainly requests to make statements more precise. Some may be only differences in opinion. I trust the authors to make the changes as needed and don’t need to see a revised manuscript.

Title: Why 3.0? Why not 2.0 or 4.0 or 3.11? If it’s a reference to *third* generation sequencing, it’s too opaque for this reviewer.

l. 42: ultra: what is ultra high-throughput sequencing as opposed to vanilla high-throughput sequencing?

l. 44: the examples are not well chosen. Long reads have not *enabled* gene cloning or introgression mapping. There are classic QTL papers by Dani Zamir about tomato introgressions. Maize TB1 was cloned the old fashioned way. The rye genome sequence assembly hasn’t contributed to climate resilient rye so far. Better examples are diploid or even tetraploid genome assembly, pan-genome projects enabled by cheap sequencing, and analysis of alternative splicing.

l. 49: marker assisted breeding doesn’t need expensive sequencing. Also see the bandwagon paper in Theoretical and Applied Genetics by Rex Bernardo on the promises and prospects of marker assisted selection (https://pubmed.ncbi.nlm.nih.gov/27681088/)

l. 53-54: I do not subscribe to a statement of such sweeping generality. In many crops or ecological model species there are still low-hanging fruits: traits controlled by single genes that haven’t been cloned yet.

l. 69-70: That sounds overly mysterious. The principle of PacBio HiFi was published in Nat Biotech.

l. 75: entire: Are there alternative methods for direct sequencing of *partial* RNA molecules?

l. 79: barrels and reader heads have not been introduced

l. 123-124 The sentence is too wordy and “last finishing” borders on the tautological. How about “Telomere-to-telomere assembly is still difficult and expensive”?

l. 128: undetected: This definition of orphan crop sounds odd to me. Some orphan crops can have large economic potential, apparent to everyone. Tef, for instance, feeds millions.

l. 129: The definition of pseudochromosome looks to me rather like the definition of “scaffold” and misses the key points that what you call pseudochromosomes (I prefer “pseudomolecules”) are in silico representatives of entire chromosome. A proposed nomenclature for them is also C-scaffold (https://doi.org/10.1093/gigascience/giz086)

l. 130 “novel”: replace it with “Assemblies generated with the latest long-read technologies”. Reference genome sequences should be preceded by “long-standing”

l. 147 check the metaphor: bottlenecks can be shifted only in the glassworks

l. 233 whose: I was misled into futile ruminations as to whether there is algal diversity whose exploration doesn’t matter all that much. Proposal: “Exploration of algal diversity at the whole genome level may underpin evolutionary research and biotechnological applications”

l. 257: Topoisomerase, DNA polymerase and condensin are functionally conserved among yeast, plants and humans. You must mean synonymous sequence conservation

l. 277 double*d* haploid is more accurate

l. 278 Even: human genomics is not years ahead in haploid genome assembly. You can visit any wheat and barley and stand a high chance of picking something effectively haploid. You need hydatiform moles to get something like haploid humans.

l. 279-80: A bit wordy. How about: Two haploid genome sequences need to be separated and assembled in a heterozygous genotype. Also note that long IBD tract can result in long runs of homozygosity in otherwise heterozygous individuals (so effective size in the diploid is less than twice the haploid).

l. 308: Gamete sequencing and AllHiC are alternative approaches worth mentioning

l. 313 I agree with the conclusion, but Mascher et al. assembled an inbred. There are potato papers to cite.

l. 317: HiFi has made genome assembly A LOT easier. It’s not a one way street, more data -> more compute muscle needed for heavy lifting. 

l. 324: The Hi-Canu and Hifiasm papers are better references

l. 326 raw data: specify what you mean by raw. Most people think of FASTQ files as Illumina raw data and spend a lot money to back them up, and are right in doing so.

l. 337 remove “almost”

l. 339 “General trend” is an overstatement. The Haghshenas reference is about an outdated approach: correction of long-reads with short-reads, the Shafin reference about ONT assembly, which you can avoid by using HiFi instead.

l. 345 This whole paragraph borders on the cliché. Nothing is unlimited in the physical world, even the resources of Google and Amazon. The establishment of bioinformatics pipelines in the cloud is beset with troubles worth mentioning such as the high cost for storage and data transfer. Either omit this paragraph, or make it more meaningful. 

l. 355 novel genomes -> species without existing reference genome sequence assemblies

l. 355 most: it’s hard to gauge the relative amounts of resequencing vs. de novo assembly, but the amount of resequencing in crop and human is also huge. I recommend to rephrase the sentence in less contentious way.

l. 368 flowcell: use a more general term, PacBio doesn’t manufacture flow cells.

l. 375 “the best plant genome” HiFiasm is also good. It would require a detailed benchmark a paper to decide with is better. I’m not aware of such a paper.

l. 376 “*if* repetitive sequences are of interest”. Given the genome sizes of most plants, lack of interest in repetitive sequence can be considered a character fault in a plant genome researcher. Even if you’re not interested in them, they will still mess up your assemblies.

l. 379-381 It does not befit this complex topic to be mentioned in passing. There are probably alternative to CAT as well.

l. 385 The key factors in PacBio sequencing are lifetime and accuracy of the polymerase. If you can double reads length (lifetime) at the cost of a minor drop in accuracy, HiFi reads may become both more accurate and longer.

l. 387 “no limit” This is like the cloud statement above. Even in practical terms, the limit is the length of the longest naturally occurring DNA polymers.

l. 390: dead end: that sounds strong and is too unspecific. Previous biotechnological inventions like PCR, cloning and SNP chips have become routine applications, rather than dead ends. It’s a valid concern, though, that the careers of those scientist that have ridden the tide of ever-improving genome assembly are in need of some refocusing.

l. 398 replace perfect by its definition: gapless telomere-to-telomere

l. 402-404: black matter etc. that’s a complicated way of referring to pan-genomes. How about: the next step after complete reference genomes are pan-genomes to capture intraspecific diversity

l. 405 Current regulation will stick by the old adage about absence of evidence. Absence of transgene insertion in the genome sequence assembly cannot prove absence of the insertion in the genome. For scientific applications, genome assemblies will good enough though.

l. 411 remove commas before PacBio and after ONT

---

## [Reviewer Report]

*Comments to Author*: The authors of Plant genome sequence assembly 3.0: progress, challenges, and future directions delivered a timely review addressing an important field of current and future research.

The review is quite focused on ONT technology with only some PacBio sprinkled in. It might be a better fit if only ONT were reviewed and if the title would reflect that. The information on PacBio is very limited in the review. Alternatively, PacBio information could be expanded where appropriate.

I have a range of suggestions that I think would improve the review since they would add more structure and I did find an error which needs correcting.

The error is in figure 1A which gives the impression that Nanopore measures current based on single bases when it indeed measures current based on approximately a k-mer (length varies based on pore type) in the channel. Please alter figure 1A to properly reflect what yields the signal.

I recommend to extend figure 2 with two more columns, one reflecting necessary instruments (if not part of the standard molecular biology lab set-up such as centrifuges) and one showing approximate cost. The review talks a lot about democratizing sequencing. It is only fair to show what the auxiliaries cost in set-up and kits.

Line by line and chapter comments as they appear in the text

l45 that claim is a bit steep, climate resilience research if a genome was delivered in the paper, please rephrase

ll49-52, please be more specific with the examples, I found myself wanting to know what was actually done; the view presented here is a bit too “bird’s eye” in my opinion.

Chapter starting at l106: I found myself wondering if a potential reader would want a bit more meat on the bones. From this paragraph a reader could not figure out what he or she would have to do to sequence and assemble a genome of interest. It would be very helpful if the expert authors could give more specifics, what can you get with Nanopore-only, what would you miss? Do you need Illumina for polishing? Do you need chromosome level assemblies (and incur the costs for additional data) for all applications? Software is only very briefly mentioned. So what kind of architecture would you need to do it in house? What kind of training would be required? Can a graduate student do it?

At the end, the paragraph claims the bottle neck shifted to DNA prep and data analysis. I suggest to restructure the chapter in the order of what one would to to get a genome: DNA isolation (and related issues, purity, length, etc.) Sequencing (what can you do with a Flongle, a Minion, a gridion and higher? What can you do with PacBio?). If ONT, basecalling (possibilities, new developments, required infrastructure). Read alignment for correction and selection. Assembly. Polishing (long reads, Illumina reads, when is it required if at all?). What kind of genome would you have now? Potential additional steps to get better or even telomere-to-telomer assemblies. I also suggest integrating information from the latter chapters into this chapter where appropriate.

l173 please define pan-genome first for readers who are not familiar with the term. One could move ll 193f up, perhaps including the “why do pangenomes” part and then explain the how. Are long read technologies equally suitable to call SNV or would you need two different sequencing technologies to get both?

ll213ff,. For the organisms listed in this chapter, it is unclear what the status of the different assemblies are. Telomer-to-telomer, serviceable, kind of serviceable or useless short read only? Many genomes are listed within the paragraph. Maybe a table which includes the organism name, genome size, assembly quality, etc. would be helpful to the reader.

l260 The idea that more genomes will get us more knowledge about genes with products of unknown function is controversial. How will sequencing more genomes get us one iota closer to knowing what the genes of unknown function are doing? I’d argue, more genome sequencing may tell us which genes are frequently occurring but will not tell us what their products do. I think, biochemistry would be more helpful than more sequences.

l316 I am not sure it is indeed a “bottle neck”. The word is overused in the review. Maybe it would be better to state the different steps required for computation and their requirements. Basecalling could be a first paragraph – what is it? What does it require in terms of compute power? More detail. Next step read alignment and correction – mention the space requirements at this step. Next step assembly. Issues here. And so on. A non-specialist will get lost when the paragraph stays structured as it is. Also consider if text here is doubled up with text in chapter Genome sequencing is accelerated, affordable, and accessible

l3341 typo interested is interest

The chapter starting at l350 is partially redundant with text above, i.e. challenges in long read sequencing. It might be useful to integrate the text in ll350-388 into the respective chapters above.

The chapter ll390-413 is partially redundant as well. One could integrate the ToL effort and others into the introduction and the paragraphs in here into their respective chapters.ll396-401 would be Genome sequencing is accelerated, affordable, and accessible, ll402-408 would be Pangenomics: From re-sequencing to reference quality genome assemblies of cultivars.

Only 409-413 How would long read sequencing give you 3-D genome architecture I am wondering. Please include a bit more information for both 3D genome structure and DNA modification so that non-experts understand the contribution of long read technologies. I am familiar with ONT for DNA methylation detection but I am not sure how PacBio may contribute to that (other than bisulfite sequencing which would not require a long read technology unless one wanted to haplophase the methylation status).

---

## [Reviewer Report]

*Comments to Author*: Dear Boas,

please accept my apologies for these late reviews, I entered this process only recently, but we could now secure three high quality reviews for your manuscript. All three reviewer are supportive of this paper, but have strong recommendations, which I believe will strengthen your work.